# Rapid riparian ecosystem recovery in low-latitudinal North China following the end-Permian mass extinction

Wenwei Guo[1,2], Li Tian[2]*, Daoliang Chu[2], Wenchao Shu[2], Michael J Benton[3], Jun Liu[4], Jinnan Tong[2]

[1]College of Culture and Tourism, Zhangzhou Institute of Technology, Zhangzhou, China; [2]State Key Laboratory of Biogeology and Environmental Geology, China University of Geosciences, Wuhan, China; [3]School of Earth Sciences, University of Bristol, Bristol, United Kingdom; [4]Institute of Vertebrate Paleontology and Paleoanthropology, Chinese Academy of Sciences, Beijing, China

*For correspondence:
tianlibgeg@cug.edu.cn

## eLife Assessment

This is a well-written **important** paper on the recovery of fauna and flora following the end-Permian extinction event in several continental sites in northern China. The **convincing** conclusion, a rapid recovery in tropical riparian ecosystems following a short phase of hostile environments and depauperate biota, is supported by an impressive amount of data from sedimentology, body fossils of animals and plants, and especially trace fossils.

**Abstract** The greatest mass extinction at the end of the Permian, ca. 252 million years ago, led to a tropical dead zone on land and sea. The speed of recovery of life has been debated, whether fast or slow, and terrestrial ecosystems are much less understood than marine. Here, we show fast reestablishment of riparian ecosystems in low-latitude North China as little as ~2 million years after the end-Permian mass extinction. The initial ichnoassemblages in shallow lacustrine and fluvial facies of late Smithian age are monospecific, devoid of infaunalization, with apparent size reduction. In the following Spathian, relatively complex, multi-level, structured riverain ecosystems had been rebuilt including medium-sized carnivores, plant stems, root traces, increased ichnological complexity, and significantly increased infaunalization. Specifically, burrowing behavior had re-emerged as a key life strategy not only to minimize stressful climatic conditions, but possibly to escape predation.

## Introduction

The end-Permian mass extinction (EPME) is one of the most devastating bio-crises in the history of life, and it destroyed both marine and terrestrial ecosystems (*Wignall, 2015*; *Fan et al., 2020*). A key feature of the EPME was the clearing of life from tropical ecosystems (e.g., *Sun et al., 2012*). Environmental extremes, especially lethal heat, largely flattened the tropical biodiversity peak, and marine animals were forced to migrate poleward or into deeper water (*Liu et al., 2020*; *Song et al., 2020*). Such effects were more brutal on land. A tropical 'tetrapod gap', spanning between 15°N and ~31°S, prevailed in the Early Triassic, or at particular intervals of intense global warming, even though the nature, temporal duration, and spatial range of the Tropical Dead Zone (TDZ) remain debated (*Bernardi et al., 2018*; *Allen et al., 2020*; *Romano et al., 2020*; *Liu et al., 2022*).

Long-term environmental perturbations after the EPME resulted in delayed recovery in the sea until the Middle Triassic (*Chen and Benton, 2012*), although several fast evolvers bounced back fast before being

**eLife digest** For over half of its history, Earth was an unwelcoming place devoid of oxygen and rich in carbon dioxide, methane and water vapor. Yet some of the earliest forms of life are thought to have emerged around 3.7 to 4.28 billion years ago, and life began to flourish during the Cambrian explosion around 500 million years ago.

Over millions of years, life was repeatedly tested and reshaped by five major mass extinctions. The largest of these, around 252 million years ago, wiped out over 95% of marine species and 70% of land or terrestrial species. Intense volcanic activity, global warming, acid rain, and elevated levels of carbon dioxide and sulfur rendered Earth a hostile environment once again for millions of years.

How long it took for environments and organisms to recover remains debated. While some marine species appear to have rebounded relatively quickly, much less is known about terrestrial species due to scarce fossil records, particularly in equatorial land basins.

To investigate if animals and plants in these regions recovered slowly or rapidly, Guo et al. analyzed rock formations from the uppermost Permian-Lower Triassic layer of the Shichuanhe, Dayulin and Liulin sections, as well as the Hongyatou, Tuncun, Mafang outcrops in the Central North China Basin.

The researchers identified newly discovered medium-sized carnivores, plant stems, and root traces. Improved ichnological criteria (methods used to study trace fossils such as burrows, tracks, and feeding marks) and significantly increased infaunalization (the extent to which organisms live within sediments) from the Heshanggou Formation of the central North China Basin suggest a relatively complex, multi-level feeding-structure within a river-associated ecosystem during the Spathian stage, just two million years after the mass extinction. This suggests that terrestrial ecosystems in equatorial regions may have recovered more rapidly and with greater complexity than previously assumed.

The study deepens our understanding of how Earth became habitable again and offers new perspectives on how life responds to extreme global warming. It suggests that, around 250 million years ago, plants and animals adapted relatively quickly to hyperthermal conditions, for example by burrowing and inhabiting riparian environments, providing valuable insights into potential biological responses to future climate change.

killed by further repeat hyperthermal events through the Early Triassic, as evidenced by the Guiyang Biota found ~1 million years (Myr) after the EPME (*Dai et al., 2023*). However, the post-extinction recovery on land has remained largely unclear. The taxonomic diversity of vertebrates may have re-flourished soon after the extinction in European Russia (*Tverdokhlebov et al., 2003*), but it apparently took longer, until the latest Early Triassic, in the Central European Basin (e.g., *Scholze et al., 2017*; *Mujal et al., 2025*). Model results of tetrapod-dominated paleocommunities from the Karoo also displayed a short-lived, unstable ecosystem during the Early Triassic *Lystrosaurus* Assemblage Zone, before being replaced by a globally stable ecological structure established in the Middle Triassic (*Roopnarine et al., 2019*; *Viglietti et al., 2022*). It seemed that plants showed a quick return in Australia (*Vajda and Kear, 2024*), yet reorganization of floral communities was hindered by repeated climatic stressors, such as the Smithian–Spathian warming event (*Mays et al., 2020*; *Vajda et al., 2020*). Likewise, the initial construction of the mesophytic flora was in the earliest Middle Triassic in North China (*Shu et al., 2022*).

Other body fossils, especially non-marine invertebrates, are relatively scarce in the post-extinction interval. The earliest Middle Triassic riverine community, consisting of insects, rare fishes, and trace fossils in equatorial western peri-Tethys (*Baucon et al., 2014*; *Mujal et al., 2017*; *Matamales-Andreu et al., 2021*) and the Anisian–Ladinian deep lake biota, comprising diverse insects, fishes, fish coprolites, and plants in tropical North China (*Zhao et al., 2020*), was thought to be representative of recovered tropical terrestrial ecosystems after the EPME. However, recent studies have revealed that biodiversity was not as low as expected in North China from the ichnological point of view, because relatively diverse trace fossils have been found in upper Lower Triassic deposits with moderate bioturbation (*Guo et al., 2019*; *Xing et al., 2021*; *Zheng et al., 2021*). Here, we report an unexpectedly rapidly recovered ecosystem from the TDZ after the EPME, based on compiled data of vertebrates, invertebrate trace fossils, and plant remains from Lower Triassic successions and outcrops in North China, to show how animals survived in harsh conditions and how the ecosystems finally recovered from the mass extinction event.

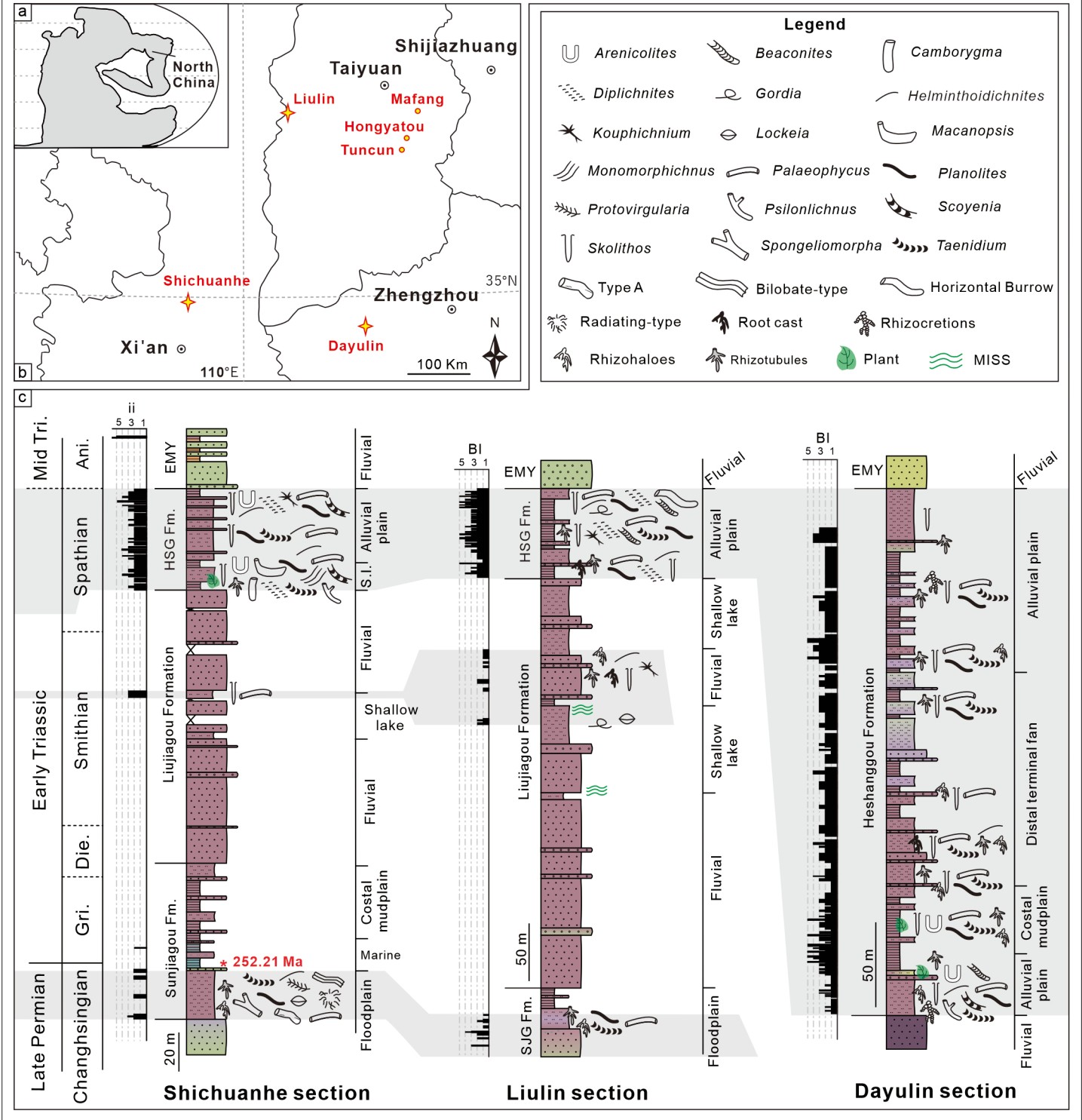

**Figure 1.** Location of the studied regions and lithological columns. (**a, b**) Permian-Triassic paleogeographic map of North China and the studied successions (stars) and outcrops (points). Base map is modified from *Sun et al., 2012*. (**c**) Depositional facies and detailed distribution of trace fossils in three main successions.

## Methods

Abundant trace fossils have been identified from the uppermost Permian–Lower Triassic of the Shichuanhe, Dayulin, Liulin sections and Hongyatou, Tuncun, Mafang outcrops in the Central North China Basin (*Figure 1*). During the early Mesozoic, the North China Block was located at about 20°N based on paleomagnetic reconstruction (*Huang et al., 2018*; *Guo et al., 2022*). The Permo-Triassic strata, typically terrestrial red-beds, comprising the Sunjiagou, Liujiagou, Heshanggou (HSG), and Ermaying formations in ascending order, display varied depositional environments from fluvial to lacustrine facies. The Sunjiagou Formation mainly consists of red massive siltstone and intercalated sandstone of varied thickness, and paleosols and trace fossils were locally developed, demonstrating floodplain to lakeshore facies (*Ji et al., 2023*; *Yu et al., 2022*). The Liujiagou Formation is characterized by thick cross-bedded sandstones and intraclast conglomerates. Lenticular sand bodies and erosive bases of sandstones are common in the lower part, indicating fluvial channel facies under braided river systems (*Zhu et al., 2019*; *Ji et al., 2023*). Interlayered thick siltstones of lakeshore facies increase upwards, accompanied by weak evidence of biological activity. The HSG is dominated by massive siltstones, with multi-layered paleosols, diverse trace fossils, and some plant remains, belonging to alluvial plain and lakeshore facies, with periodic aerial exposure (*Zhu et al., 2020*; *Ji et al., 2023*).

Recently, a U-Pb CA-ID-TIMS age calibrated magnetostratigraphy from the Shichuanhe section provided a basic geochronological timescale for the fossil-poor strata in North China (*Guo et al., 2022*; *Supplementary file 1*). Accordingly, the Permian–Triassic Boundary is constrained in the upper part of the Sunjiagou Formation based on the age of 252.21±0.15 million years ago (Ma), with the base of the Smithian and Spathian being roughly located in the lower and upper Liujiagou Formation, respectively. However, loss of several magnetozones in a regional hiatus makes it difficult to place the Lower–Middle Triassic Boundary precisely, which was tentatively put at the lithological contact of the HSG and the overlying Ermaying Formation (*Guo et al., 2022*). The 245–247 Ma ages from tuff layers at the base of the Ermaying Formation roughly support this correlation (*Zhu et al., 2022*).

In order to discriminate the recovery stages, several ichno-ecological criteria are used. Both bioturbation index (BI) and ichnofabric index (ii), which have been critically reviewed by *Luo et al., 2020*, are used to quantify bioturbation intensity. Values of ii and BI, ranging from 1 to 6 and 0 to 6, respectively, co-indicate the gradual increase of biotic disturbance from no bioturbation to total homogenization of sediments. Ichnodiversity represents the number of ichnotaxa, but it is not strictly equivalent to biodiversity, as a certain trace can be made by different animals and multiple trace types can originate from a single taxon (*Luo et al., 2020*). Ichnodisparity emphasizes the variability of architectural designs of trace fossils (*Buatois et al., 2017*), morphologically distinct forms being termed ichnomorphs. Therefore, both ichnodiversity and ichnodisparity are employed to assess the behavioral responses of animals and the stage of infaunal biotic recovery. Size and penetration depth are also measured in places that can represent the average level of each burrow. Tiering, referring to the life position of an animal vertically in the sediment, is divided into surficial, semi-infaunal (0–0.5 cm), shallow (0.5–6 cm), intermediate (6–12 cm), and deep infaunal tiers (>12 cm), based on *Minter et al., 2017*.

Thin sections of plant stem specimens were prepared to examine vertical and cross-sectional microstructures. Meanwhile, Micro-CT scanning (SkyScan 1172 X-ray; State Key Laboratory of Biogeology and Environmental Geology) was employed to reconstruct the internal structures of stems.

## Results and discussion

### Infaunal crisis and living strategy after the EPME in North China

An infaunal crisis, marked by the disappearance of the moderately diversified pre-extinction ichnofauna and the absence of biogenic structures, is identified in successions of late Changhsingian–early Smithian age. The latest Permian floodplain facies were mainly occupied by shallow–intermediate tiers of freely motile non-specialized deposit-feeding animals (*Figure 1*), akin to the Paleozoic suites reported before (*Minter et al., 2017*). Both ichnodiversity and ichnodisparity decline abruptly in the middle Sunjiagou Formation (near the 252.21 Ma aged tuff layer), following a prolonged non-bioturbated interval (*Guo et al., 2019*; *Guo et al., 2022*; *Xing et al., 2021*). The infaunal crisis was contemporary with the extirpation of the pareiasaur fauna (*Shihtienfenia*) and deforestation of the youngest Palaeozoic *Ullmannia-Pseudovoltzia-Germaropteris* assemblage in North China, representing the EPME on land (*Liu et al., 2022*; *Shu et al., 2022*).

The early recovery ichnofauna preserved in the upper Liujiagou Formation, about late Smithian in age, was short-lived. This lakeshore ichnofauna, including seven ichnogenera of six ichnomorphs, is characterized by surficial and semi-infaunal tiered simple burrows or trails, with rare trackways and root traces, which weakly disturbed the sediments. Dwarfism in trace makers is observed from all ichnogenera, with burrow sizes reduced from a mean of 4.06 mm before the EPME (n=779)–2.06 mm (n=341) in the late Smithian (Figure 4), and as seen in individual ichnogenera such as *Kouphichnium* (*Shu et al., 2018*). The depauperate ichnofauna of the late Smithian was monospecific, representing initial recolonization of empty niches by opportunists. However, recurrent occurrences of microbially induced sedimentary structures (MISS) in the Liujiagou Formation show that depressed ecosystems persisted into the Smithian (*Tu et al., 2016*; *Chu et al., 2017*). Earlier work has shown that increases in microbial abundance were generally associated with hyperthermal events, the principal cause for mass extinction on land (*Mays et al., 2021*). Accumulations of microbes were favored by low dissolved oxygen concentration conditions, and their secondary metabolites could also be toxic to animals (*Pacton et al., 2011*; *Paerl and Otten, 2013*). Therefore, repeated thriving of MISS during the Dienerian–Smithian interval disrupted ecological stability in freshwater ecosystems and delayed biotic recovery.

Abundant trace fossils are identified in the Spathian HSG Formation, comprising 16 ichnogenera of nine ichnomorphs and two informally designated types (*Figure 2*). This morphologically complex ichnofauna was dominated by actively filled burrows, with few arthropod trackways, which were probably produced by decapod crustaceans, myriapods, and insects. The lakeshore and alluvial plain facies were occupied by multi-tiered traces, including surficial trackways (e.g., *Diplichnites*; *Kouphichnium*), semi-infaunal (e.g., *Helminthoidichnites*; *Gordia*), shallow (e.g., *Palaeophycus*; *Scoyenia*), intermediate (e.g., *Taenidium*), and deep tiers (*Camborygma*; *Skolithos*; *Figure 1c*). Trace producers that colonized varied ecospace and shallower tiers are generally crosscut by deeper penetrative burrows (*Figure 2a*), resulting in moderately to substantially bioturbated deposits, with ii 2–3 and BI 3–4 at most layers. In several horizons, the uppermost few centimeters of sediments are totally obliterated, mostly by the activity of deposit feeding animals. However, distal terminal fan facies are weakly reworked by simple traces such as *Skolithos* and *Palaeophycus* or root traces. Additionally, average burrow sizes of all ichnogenera also increased to 3.9 mm in the HSG (n=2241, Figure 4).

Several large burrows in the Spathian indicate the occurrence of advanced ecosystem engineers. *Camborygma litonomos* were found in the basal HSG of the Shichuanhe section and outcrop in Hongyatou, co-occurring with in situ preserved *Neocalamites* plant fossils, and other traces, such as *Diplichnites*, *Monomorphichnus,* and *Skolithos*. Surfaces of *C. litonomos* were occasionally intertwined with rhizotubules (*Figure 2l*), suggesting that those plants could be important constituents in the diet of crayfish or that the crayfish might have hidden among the roots. In addition, another large burrow was identified in the Liulin section, this sub-horizontal unbranched burrow displays probable longitudinal scratch marks on the external surface, but poor preservation and limited specimens hinder a definite designation. ?*Beaconites*, found in the lower part of the HSG at Mafang, are characterized by meniscate structures (*Figure 2o and p*), akin to those of large *Beaconites* from breccia facies in the Devonian red sandstone of Britain (*Brück, 1987*). Although the biological nature of these large burrows cannot be confirmed, their activities increased biogenic reworking of sediment and soils, improved geochemical recycling and ecosystem complexity in the Spathian, implying key roles in ecosystem functioning.

Although climatic and environmental conditions in the early Spathian were still not cool and wet enough for thriving and abundant life, a fossorial strategy would have been useful for terrestrial animals to avoid heat and aridity. Midday temperatures >35°C, as occurred during peaks of global warming at the EPME and at points through the Early Triassic, cannot be tolerated for long by terrestrial (or aquatic) animals (*Benton, 2018*; *Liu et al., 2022*). The increase in tetrapod burrow abundance and complexity in the Lower Triassic suggests that a fossorial lifestyle allowed tetrapods to endure harsh post-extinction environmental conditions (*Marchetti et al., 2024*). Likewise, we envisage that infaunalization could also have been a vital strategy for invertebrates to survive and thrive. More intensively occupied ecospace in the late Spathian, characterized by increased burrowing and complicated crosscutting relations among ichnogenera, may indicate an adaptive response to heightened predation pressure or competition for available resources.

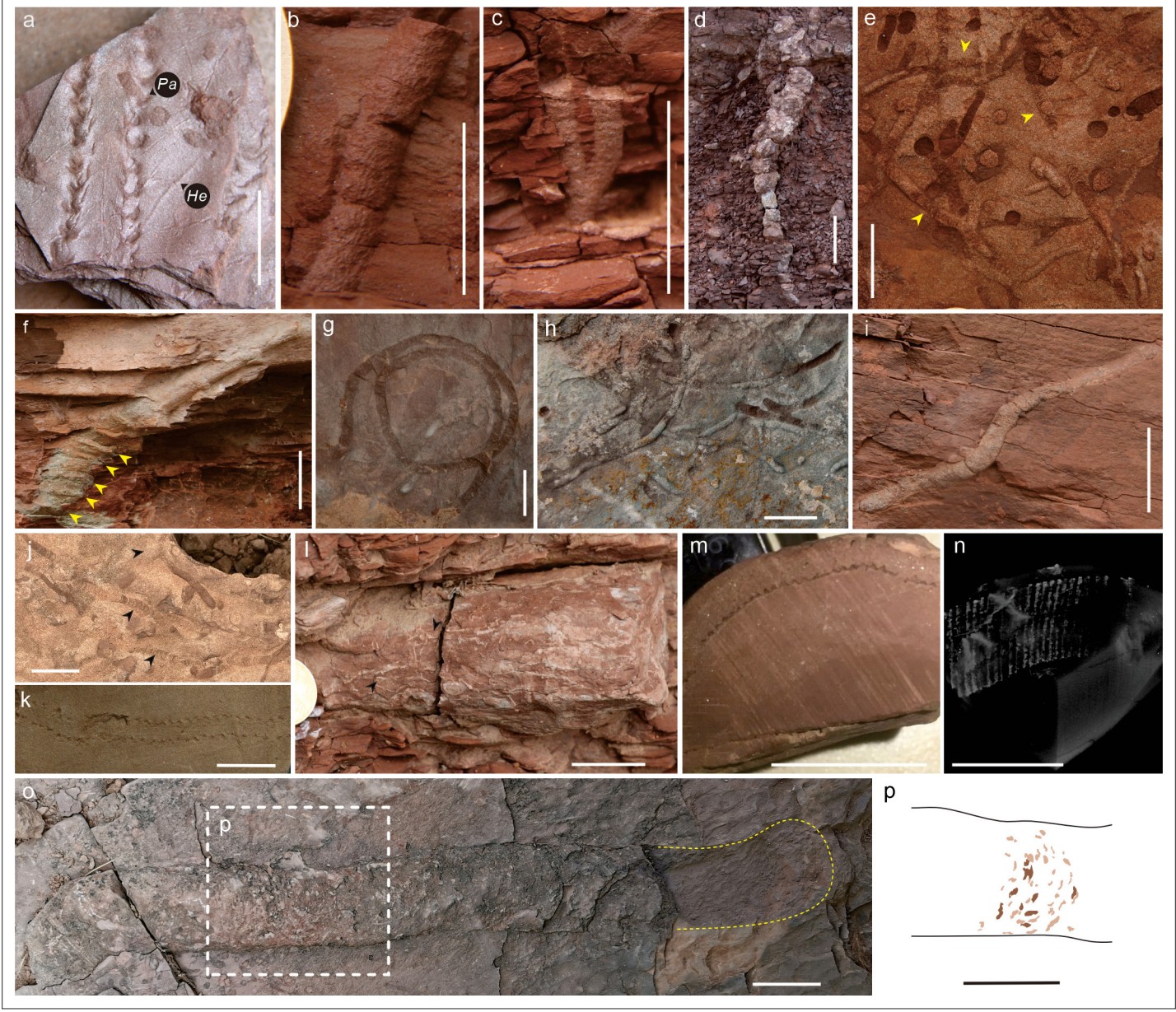

**Figure 2.** Ichnofossils from the Heshanggou Formation of North China. (**a**) Shallow tiers *Kouphichnium* and *Helminthoidichnites* (He) are crosscut by immediate tier *Palaeophycus* (Pa). (**b**) *Skolithos* cf. *serratus* with faint oblique striations. (**c**) Y-shaped *Psilonichnus* isp. (**d**) Downward unbranched and tapered rhizocretion. (**e**) Shallow tiers of *Beaconites coronus*, arrows show tightly stacked arcuate meniscus. (**f**) *Camborygma* isp. shows enlarged terminal chamber and possible transverse scratches (arrows). (**g**) *Gordia* isp. with darker and finer infills. (**h**) High-density *Palaeophycus tubularis* preserved on the sole of thick sandstone. (**i**) Inclined *Planolites beverleyensis* within siltstone. (**j**) *Taenidium barretti* (arrows) pass through the rippled surface. (**k**) Horizontal *Camborygma*, the outer surface is intertwined with tiny root traces (arrows). (**l**) Biserial *Diplichnites gouldi*. (**m, n**) Internode cross-section of *Neocalamites* stem and micro-CT structure, showing the clear ribs and grooves. (**o, p**) large? *Beaconites* on top of rippled sandstone, rectangle in (**p**) shows meniscus-like portions comprising gritty infillings. Scale bars of (**d, f, h, o, p**) are 40 mm, the rest are all 20 mm.

## Fast-recovered terrestrial ecosystem in tropical region

Our results also shed light on the timing of the TDZ. The late Smithian-age ichnofauna, although impoverished, represents early opportunist-dominated communities that explore empty ecospace under inhospitable environments, which constrain persistence of the TDZ to the late Smithian in North China. Newly discovered tetrapods from the HSG provide crucial insights into Spathian-aged ecosystems. Historically, vertebrates found in this lithological unit were exclusively from its middle–upper portions (*Li et al., 2008*; *Figures 3 and 4*). The presence of Archosauromorpha (e.g., *Fugusuchus*;

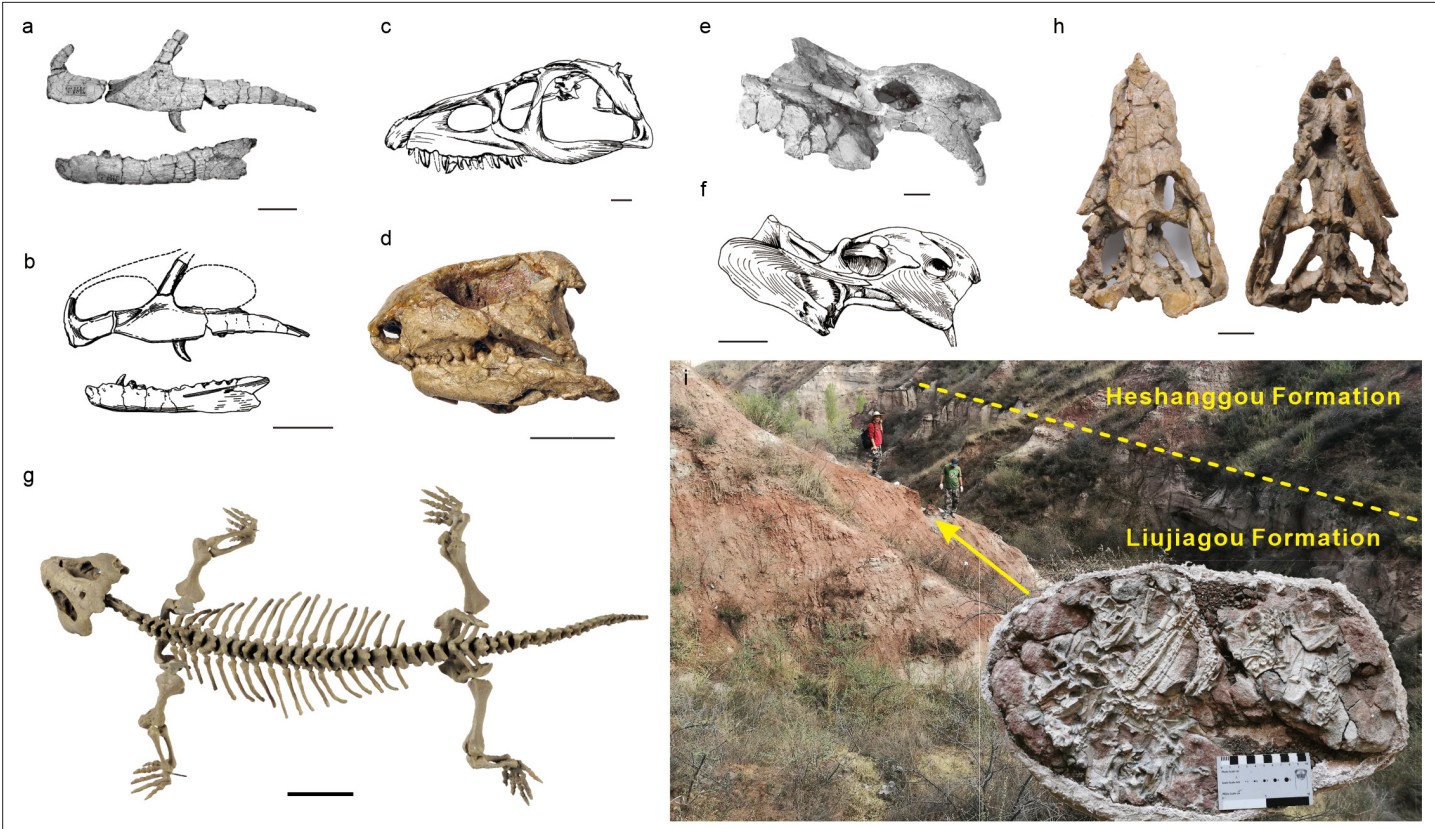

**Figure 3.** Vertebrates from the Heshanggou Formation of North China. (**a, b**) Skull elements of *Xilousuchus sapingensis* and drawings. (**c**) *Fugusuchus hejiapanensis*. (**d**) *Eumetabolodon bathycephalus*. (**e, f**) *Shaanbeikannemeyeria xilouensis* and drawing. (**g**) *Pentaedrusaurus ordosianus*. (**h**) *Hazhenia concava*, (**i**) Dashed line shows the boundary between the Liujiagou and Heshanggou formations. Arrow displays fossil horizon of the inserted picture at the base of the Heshanggou Formation. (**b, c, f**) are from *Li et al., 2008*. Scale bars are 40 mm.

*Figure 3c*) and Procolophonomorpha (e.g., *Eumetabolodon*; *Figure 3d*) could reflect their tolerance to hot and arid conditions, while the initial diversification of archosauromorphs in the Olenekian was interpreted as a response to empty ecological space after the EPME (*McLoughlin et al., 2020*; *Romano et al., 2020*). Herein, a cluster of tetrapod skeletons, including a few articulated bones, was found near the base of HSG (early Spathian; *Figure 3i*). Body trunk lengths are estimated at 30–40 cm for the vertebrates, and the postcranial skeleton suggests a carnivorous feeding strategy. Although the specimens are not yet fully prepared for taxonomic description, they clearly show the existence of tetrapods at this level, narrowing the 'tetrapod gap' to the Spathian.

Plants, as key components of the ecosystem, were patchily distributed in the Early Triassic (*Shu et al., 2022*). However, root traces (rhizoliths) are quite abundant, especially in the Dayulin section (*Figure 2d*). The different styles of preservation and various morphologies of rhizoliths provide information about the moderately to relatively well-drained red paleosols in alluvial plain facies (*Kraus and Hasiotis, 2006*). Specifically, in situ preserved vertical *Neocalamites* stems and *Pleuromeia* found in the lower HSG of the Shichuanhe and Hongyatou sections both occur in tandem with crayfish trace fossils and other burrows.

Delayed terrestrial recovery was proposed based on the Tongchuan fauna from North China, which consisted of diverse insects, ostracods, fishes, etc., signaling a fully recovered deep lake ecosystem in the Middle Triassic, that is, ~8–12 Myr after the EPME (*Zheng et al., 2018*; *Zhao et al., 2020*). However, compiled paleontological data herein show that ecosystems on land occurred in tropical regions as early as in the Spathian, ~2 Myr after the EPME. The enhanced floral coverage attributed to different types of roots and plant fossils had great impact on the initiation of a post-EPME ecosystem,

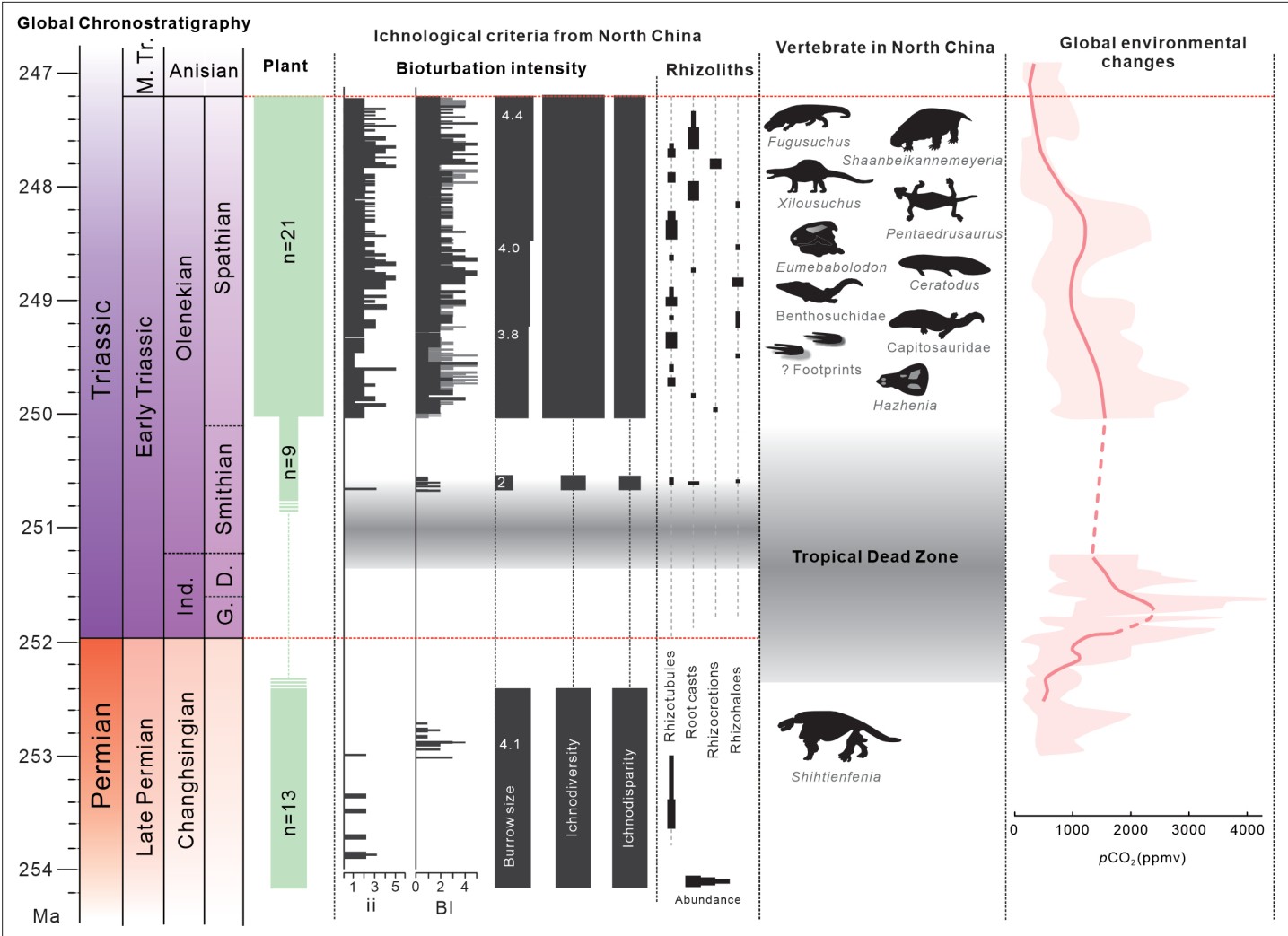

**Figure 4.** Ichnofossil data in North China and global terrestrial ecosystem changes from latest Permian to earliest Middle Triassic. Geochronological timescale is based on the latest version of the International Chronostratigraphic Chart (https://stratigraphy.org/). Plant richness from *Shu et al., 2022*. Numbers in the Burrow size column represent the mean trace fossil sizes from the investigated interval. Drainage conditions of paleosols are inferred from preservation of rhizoliths and their relative depths. Ranges of microbially induced sedimentary structures (MISS) are from *Chu et al., 2017*. Atmospheric $CO_2$ curves are modified from *Joachimski et al., 2022*. Ind., Induan; G., Griesbachian; D., Dienerian.

especially the co-occurring stems and trace fossils, which suggest that riverain realms might have been refugia for the survival and evolution of the Spathian biota (*Figure 5*). Increased abundance and quantity of faunal communities can be inferred from ichnodiversity, with possible candidate trace producers, including limuloids, crayfishes, spinicaudatans, insects, and even small-sized vertebrates, and moderate bioturbation made by high-density resting traces, respectively. Coeval tetrapods, despite being rare, do show the existence of carnivores and further complexity in local ecological structures. Reconstruction of freshwater ecosystems in the Spathian was also facilitated by ameliora-tion of the climate (*Figure 4*). Paleosol-based paleoclimatic reconstructions suggest that precipitation was ~520–680 mm/year in the late Spathian, with $pCO_2$ estimated from paleosols at 1523±417 ppm (*Joachimski et al., 2022*; *Yu et al., 2022*), indicating mitigation of hyperthermal conditions. Geochem-ical proxies of weathering intensity, salinity, and clayiness, along with increased hygrophyte/xerophyte ratio, also demonstrate the transition to wetter conditions (*Shu et al., 2022*; *Zhu et al., 2022*).

## Conclusions

The diverse community of the HSG Formation, consisting of tetrapods, plant stems, rhizoliths, and diverse ichnofossils, suggests a rapid recovery of life in low-latitude terrestrial environments in the

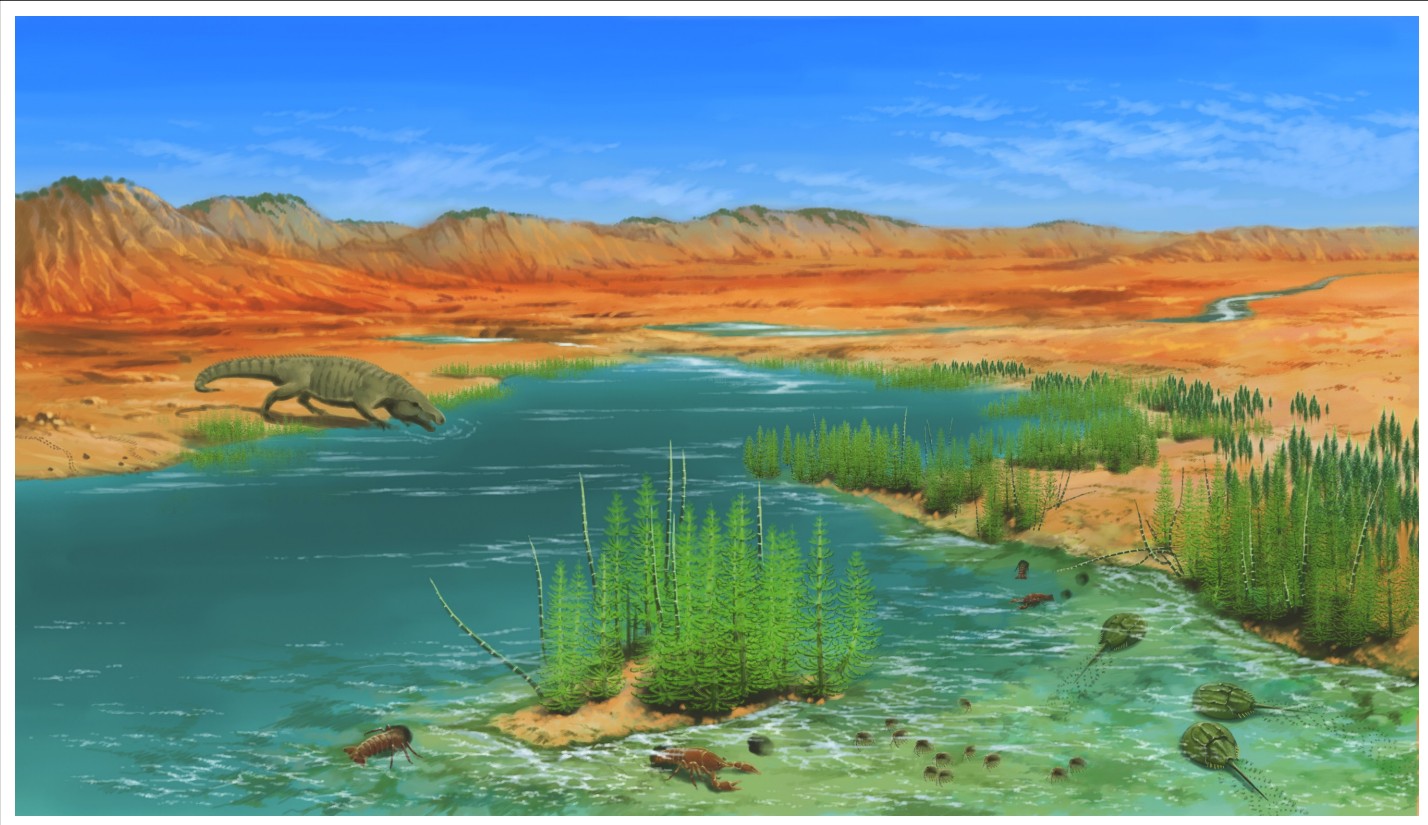

**Figure 5.** Reconstruction of the Spathian (Heshanggou Formation) coastal mudplain to alluvial ecosystem in North China. Plant communities in the coastal mudplain and alluvial facies are depauperate, dominated by *Neocalamites* and *Pleuromeia*, and only diverse at the top of the Heshanggou Formation (late Spathian; *Shu et al., 2022*). Fossil plants and tetrapods, coupled with diverse invertebrates, including limuloids, crayfish, spinicaudatans, and insects, etc., reveal reorganization of a relatively complex ecosystem in riverain regions during the Spathian.

The artistic illustration was credited by J. Sun.

Spathian, as early as ~2 Myr after the end-Permian biotic crisis. The newly discovered vertebrate fossils are medium-sized carnivores of approximately early Spathian age, representing the earliest Triassic-aged tetrapods found in North China. High ichnodiversity and ichnodisparity, and three types of large burrows, not only result in intensified bioturbation but provide additional information about the enriched local biota. Furthermore, enhanced burrowing behavior is considered a key survival-recovery strategy to adapt to harsh climatic and environmental conditions on land. The co-occurring trace fossils and stems, and coeval tetrapod in alluvial plain facies, suggest that the riverain regions could be refugia for the reorganization of post-EPME ecosystem.

## Acknowledgements

We thank Kaixuan Ji, Gan Liu, and Yuyang Wu for assistance in the field. We also thank anonymous reviewers for their comments and constructive suggestions. This work was supported by the National Natural Science Foundation of China (grant nos. 42030513) to LT, DC, and JT, the Strategic Priority Research Program of the Chinese Academy of Sciences (grant no. XDB26000000) to JL and the Natural Environment Research Council (UK) of grant no. NE/P013724/1 to MJB.

# Additional information

## Funding

| Funder | Grant reference number | Author |
| --- | --- | --- |
| National Natural Science Foundation of China | 42030513 | Li Tian<br>Daoliang Chu<br>Jinnan Tong |
| Chinese Academy of Sciences | XDB26000000 | Jun Liu |
| Natural Environment Research Council | NE/P013724/1 | Michael J Benton |

The funders had no role in study design, data collection and interpretation, or the decision to submit the work for publication.

## Author contributions

Wenwei Guo, Conceptualization, Resources, Data curation, Software, Formal analysis, Validation, Investigation, Visualization, Methodology, Writing – original draft; Li Tian, Conceptualization, Resources, Data curation, Formal analysis, Supervision, Funding acquisition, Validation, Investigation, Visualization, Methodology, Writing – original draft, Project administration, Writing – review and editing; Daoliang Chu, Resources, Data curation, Investigation, Methodology, Writing – review and editing; Wenchao Shu, Data curation, Formal analysis, Investigation, Writing – review and editing; Michael J Benton, Supervision, Project administration, Writing – review and editing; Jun Liu, Data curation, Formal analysis, Writing – review and editing; Jinnan Tong, Supervision, Funding acquisition, Project administration, Writing – review and editing

## Author ORCIDs

Wenwei Guo (ID) https://orcid.org/0000-0002-9933-3264
Li Tian (ID) https://orcid.org/0000-0003-3005-2007
Michael J Benton (ID) https://orcid.org/0000-0002-4323-1824

Reviewer #1 (Public review): https://doi.org/10.7554/eLife.104205.4.sa1
Reviewer #2 (Public review): https://doi.org/10.7554/eLife.104205.4.sa2
Reviewer #3 (Public review): https://doi.org/10.7554/eLife.104205.4.sa3
Author response https://doi.org/10.7554/eLife.104205.4.sa4

---

# Additional files

## Supplementary files
MDAR checklist

Supplementary file 1. Integrated stratigraphic timescale and depositional environment.

## Data availability
All data generated or analyzed during this study are included in the manuscript and supporting files.

---

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
