## [Editor Report · eLife Assessment]

This is a well-written **important** paper on the recovery of fauna and flora following the end-Permian extinction event in several continental sites in northern China. The **convincing** conclusion, a rapid recovery in tropical riparian ecosystems following a short phase of hostile environments and depauperate biota, is supported by an impressive amount of data from sedimentology, body fossils of animals and plants, and especially trace fossils.

---

## [Referee Report · Reviewer #1 (Public review)]

Summary:

This is a very well-written paper presenting interesting findings related to the recovery following the end-Permian event in continental settings, from N China. The finding is timely as the topic is actively discussed in the scientific community. The data provides additional insights into the faunal, and partly, floral global recovery following the EPE, adding to the global picture.

Strengths:

The conclusions are supported by an impressive amount of sedimentological and paleontological data (mainly trace fossils) and illustrations.

---

## [Referee Report · Reviewer #2 (Public review)]

Summary:

The authors made a thorough revision of the manuscript, strengthening the message. They also considered all the comments made by the reviewers and provided appropriate and convincing arguments.

Strengths:

The revised manuscript clarifies all the major points raised by the reviewers, and the way the information is presented (in the text, figures and tables) is clear.

Weaknesses:

The authors provided an appropriate and convincing rebuttal regarding the potential weakness I pointed out in the first review of the manuscript. Therefore, I do not see any major issue in their work.

---

## [Referee Report · Reviewer #3 (Public review)]

Summary:

The manuscript by Guo and colleagues features the documentation and interpretation of three successions of continental to marginal marine deposits spanning the P/T transition and their respective ichnofaunas. Based on these new data inferences concerning end-Permian mass extinction and Triassic recovery in the tropical realm are discussed.

Strengths:

The manuscript is well written and organized and includes a large amount of new lithological and ichnological data that illuminate ecosystem evolution in a time of large scale transition. The lithological documentations, facies interpretations and ichnotaxonomic assignments look alright (with few exceptions).

---

## [Author Response]

The following is the authors’ response to the previous reviews

**Public Reviews:**

**Reviewer #1 (Public review):**
Summary:This is a very well-written paper presenting interesting findings related to the recovery following the end-Permian event in continental settings, from N China. The finding is timely as the topic is actively discussed in the scientific community. The data provides additional insights into the faunal, and partly, floral global recovery following the EPE, adding to the global picture.Strengths:The conclusions are supported by an impressive amount of sedimentological and paleontological data (mainly trace fossils) and illustrations.

We thank Reviewer #1 for the positive assessments.

Weaknesses: [eliminated in revision]

We thank Reviewer #1.

**Reviewer #2 (Public review):**
Summary:The authors made a thorough revision of the manuscript, strengthening the message. They also considered all the comments made by the reviewers and provided appropriate and convincing arguments.Strengths:The revised manuscript clarifies all the major points raised by the reviewers, and the way the information is presented (in the text, figures and tables) is clear.

We thank Reviewer #2 for the positive comments on our work.

Weaknesses:The authors provided an appropriate and convincing rebuttal regarding the potential weakness I pointed out in the first review of the manuscript. Therefore, I do not see any major issue in their work.Introduction(1) P. 2, L. 32: Replace "to migrated" with "to migrate".

Revised as suggested.

(2) P. 3, L. 43-44: We recently published a review article on the tetrapod terrestrial record from the Central European Basin, showing that Olenekian tetrapod faunas (and ichnofaunas) were already quite rich and diverse. Article: https://doi.org/10.1016/j.earscirev.2025.105085

Yes, we have read this paper. This summary is very important for the understanding of the biotic recovery after the PTME, especially in the early stage. We have added the new result in our manuscript.

(3) P. 3, L. 57: Replace "recovered terrestrial ecosystems in tropical" with "recovered tropical terrestrial ecosystems".

Revised as suggested.

Results and Discussion(4) P. 6, L. 118: Replace "declined" with "decline".

Revised as suggested.

(5) P. 7, L. 131: Replace "microbial" with "microbially".

Revised as suggested.

Conclusions(6) P. 11, L. 224: Replace "as little as" with "as early as".

Revised as suggested.

(7) P. 11, L. 227: Replace "not only results in" with "not only result in".

Revised as suggested.

(8) 11, L. 230: Replace "suggesting" with "suggest".

Revised as suggested.

**Reviewer #3 (Public review):**
Summary:This manuscript by Guo and colleagues features the documentation and interpretation of three successions of continental to marginal marine deposits spanning the P/T transition and their respective ichnofaunas. Based on these new data inferences concerning end-Permian mass extinction and Triassic recovery in the tropical realm are discussed.Strengths:The manuscript is well-written and organized and includes a large amount of new lithological and ichnological data that illuminate ecosystem evolution in a time of large-scale transition. The lithological documentations, facies interpretations, and ichnotaxonomic assignments look okay (with a few exceptions).

We thank Reviewer #3 for the positive assessments.

Weaknesses:Weaknesses: [all eliminated in revision]

We thank Reviewer #3.